# COVID-19 Vaccine Hesitancy among Parents of Children with Chronic Liver Diseases

**DOI:** 10.3390/vaccines10122094

**Published:** 2022-12-07

**Authors:** Sally Waheed Elkhadry, Tahany Abd El Hameed Salem, Abdelhamid Elshabrawy, Shymaa Sami Goda, Howyda Ali Al Bahwashy, Naglaa Youssef, Mai Hussein, Ramy Mohamed Ghazy

**Affiliations:** 1Department of Epidemiology and Preventive Medicine, National Liver Institute, Menoufia University, Menoufia 32511, Egypt; 2Department of Pediatric Hepatology, Gastroenterology, and Nutrition, National Liver Institute, Menoufia University, Menoufia 32511, Egypt; 3Department of Bio Statistics and Demography, Faculty of Graduate Studies for Statistical Research, Cairo University, Cairo 12556, Egypt; 4Department of Medical-Surgical Nursing, College of Nursing, Princess Nourah bint Abdulrahman University, Riyadh 11671, Saudi Arabia; 5Alexandria Clinical Research Administration, Health Affairs Directorate, Ministry of Health and Population, Alexandria 21554, Egypt; 6Tropical Health Department, High Institute of Public Health, Alexandria University, Alexandria 21561, Egypt

**Keywords:** COVID-19, chronic liver disease, parental attitude toward childhood vaccination, vaccine hesitancy, vaccine trust, vaccine effectiveness

## Abstract

Children with chronic medical conditions are more susceptible to developing a serious negative outcome from corona virus disease 2019 (COVID-19) than healthy children. This study investigated the extent of COVID-19 vaccine hesitancy (VH) and its predictors in parents of children with chronic liver disease (CLD) in Egypt. Methods: A cross-sectional study was conducted at the National Liver Institute from September to October 2022, using a random sampling method. Data were collected using the validated Arabic version of parents’ attitudes about childhood vaccines (PACV) scale. Structural equation modeling (SEM) and discriminant analysis were used to identify direct and indirect determinants of VH. Results: Of the 173 participating parents, 81.5% hesitated to vaccinate their child. Relevant characteristics for hesitancy included being the mother of the child (88.2%), younger than 40 years (92.9%), illiterate (92%), unemployed (88.8%), without health insurance (87.8%), unvaccinated against COVID-19 (97.2%), refused to complete vaccinations (85.7%), and not having chronic disease (85.7%) (*p* < 0.05). Previous COVID-19 infection of children motivated vaccination (*p* < 0.0001). Median total PACV, attitude, and trust scores were significantly higher in the hesitant group than the vaccinated group (*p* = 0.023). SEM suggests that child age and family size have a direct effect, while education level, and income have indirect effects on parents’ hesitancy. The model showed acceptable goodness of fit (GFI = 0.994, CFI = 1, RMSEA < 0.0001). A 92.9% corrected classification of the discriminator VH variables was determined using the discriminant analysis model (safety and efficacy, attitude and trust, child age, and family size). Conclusions: Many socioeconomic factors significantly affect parents’ attitudes toward their child’s vaccination. Thus, increasing parents’ awareness of the importance of childhood vaccination, especially among this risky group, may enhance their decision-making ability regarding vaccinating their children.

## 1. Introduction

Coronavirus 2019 (COVID-19) is an emerging infectious disease caused by the severe acute respiratory syndrome coronavirus 2 (SARS-CoV-2). As of 3 November 2022, the World Health Organization (WHO) estimated that the global COVID-19 pandemic has infected approximately 628 million subjects worldwide, resulting in 6.6 million deaths [1]. Global epidemiological data for COVID-19 among children and adolescents is insufficient, which makes it difficult to estimate the impact of the disease on children [2,3]. Globally, the prevalence of sever acute respiratory syndrome corona virus 2 (SARS-CoV-2) infection, the causative agent of COVID-19 among those aged less than 20 years, ranges from 0% to 36% of all reported cases. This prevalence is higher among adolescents aged 10–19 years than among children aged 0–9 years (62% vs. 38%, respectively) [2]. Egypt had reported approximately 515.4 thousand confirmed cases, with 24.8 thousand deaths from the start of the pandemic by 3 November 2022 [4]. Approximately 52% of Egyptians are aged ≤24 years, and this cohort is considered the most susceptible to the social and economic effects of COVID-19 [5,6].

The impact of SARS-CoV-2 infection is mild or moderate for most children [7]. While infection can cause serious illness or death in children, it is impossible to anticipate who will become mildly or severely sick [8]. Approximately 0.4% of deaths due to COVID-19 have been reported among children aged <20 years (53% among children aged 10–19 years and 47% among children aged 0–9 years) [9]. Children with comorbidities show an elevated risk of hospitalization and COVID-19-related severe morbidity and mortality [7,8,9,10,11]. These comorbidities have a remarkable effect on mortality among these age groups as well, alongside strained health systems, household income loss, disruptions to care-seeking, and omitted preventive measures, such as vaccination [9].

Vaccines have been approved for use in children and adolescents to prevent adverse consequences of SARA-CoV-2 infection. Vaccination can give children increased confidence in participating in childcare, school, and other activities, which can make families feel less stressed [8,12]. The Centers for Disease and Control (CDC) and WHO have recently recommended vaccination for everyone, even children older than 6 months of age, and booster doses in children aged >5 years. Pfizer/BioNTech and Moderna COVID-19 vaccines are recommended for people aged 6 months and older [13,14] and Novavax for people aged 12 years or older [15]. The first dose of a COVID-19 vaccine offers protection against becoming very ill. Booster doses provide stronger and longer-lasting protection against COVID-19 variants [7]. A total of 100.3 million vaccine doses had been administered as of 30 October 2022 [4]. In November 2021, the Egyptian government approved Pfizer’s COVID-19 vaccine for children between the ages of 12 and 18 years [16].

Chronic liver disease (CLD) is a progressive deterioration of liver functions lasting more than 6 months [17]. CLD is the continuous inflammation, destruction, and regeneration of the liver parenchyma that can result in irreversible cirrhosis. The spectrum of CLD etiologies is broad, including toxins, infection, autoimmune diseases, genetic, and metabolic disorders. Children with CLD are at high risk of contracting COVID-19 and tend to have a more severe disease course [18]. In Egypt, approximately 582,638 children below 14 years old have CLD [19]. A recent cross-sectional study of 349 children with cholestatic liver disease in Egypt found an incidence of SARA-CoV-2 of 8%, and 43% of these experienced significantly higher rates of hepatic complications and showed increased need for medical care services [18].

Consequently, COVID-19 vaccination and other public health strategies should consider the risks that children with health conditions such as CLDs might experience [10]. Population hesitancy to be immunized negatively affects the vaccination programs’ success [20]. The WHO defines vaccine hesitancy (VH) as “delay in acceptance or refusal of vaccination despite the availability of vaccination services.” [21]. COVID-19 vaccination hesitancy is associated with substantially more resistance than previous vaccines [22]. Many studies have reported high rates of VH in the region towards either the initial series of vaccination [23,24,25,26] or the booster dose [27,28]. However, there is scarcity in the literature regarding the parent’s attitude concerning children’s vaccination, especially among parents of children with chronic diseases. Having a child with a chronic disease may negatively affect parental decision-making regarding vaccination. Given that, this study aimed to investigate COVID-19 VH and its predictors among parents of children with CLD in Egypt.

## 2. Materials and Methods

### 2.1. Design and Setting

This cross-sectional study was conducted at outpatient clinics of the National Liver Institute (NLI), Egypt, from September to October 2022. Trained pediatricians conducted face-to-face interviews with parents of children with CLD.

### 2.2. Population, Sample Size, and Sampling Technique

The NLI receives monthly approximately 500 children suffering from various liver diseases (LD) for follow-up, outpatient, and inpatient healthcare services. Parents with at least one child (aged 5–18 years old) with CLD were the target population. Approximately 19.1% of parents in low-middle countries are hesitant to vaccinate their children with CLD against COVID-19 [29]. The sample size was calculated with a 5% margin of error, a 95% confidence interval (CI), and the design effect for the cluster surveys (DEFF) was equal to one. The estimated minimum sample was 162 participants [30]. The sample was increased to 173 to compensate for incomplete data. Participants were recruited using systematic random sampling. The total sampling frame of N = 500 children attending NLI was divided by our sample size *n.* = 170 to obtain the systematic sampling interval (k = N/*n.* = 3). The first case was selected using simple random selection, followed by every third CLD child.

### 2.3. Data Collection Questionnaire

The questionnaire included three sections. Section 1 collected parents’ sociodemographic characteristics and COVID-19-related data (i.e., parents’ age, sex, education, number of children, relation to the child (mother or father), place of work (i.e., government, private, or unemployed), health insurance coverage, income level (i.e., not enough, on a loan and but cannot pay back; not enough, on a loan but can pay it back; enough; or enough and saving), history of parents’ COVID-19 infection, parents’ vaccination, and parents’ medical history). Section 2 included the children’s characteristics (i.e., child age, sex, prior vaccination history, and history of chronic diseases). Section 3 included a valid Arabic version of the Parent Attitudes about Childhood Vaccines (PACV), a reliable scale for use in predicting parents’ intention to vaccinate [31]. PACV has three main domains: (i) behavior, (ii) attitude, and (iii) safety and efficacy [32] (Appendix A). 

### 2.4. Ethical Considerations

This study was approved by the ethics committee of the NLI (IRB number 0032/2022). All parents provided written informed consent before participation. The parents gave their written informed consent after the purpose of study was explained. The parents were informed that their participation is voluntary and they can refuse participation or withdraw their consent without any effect on their children care or provided healthcare services. Confidentiality of the parents and their children were maintained by coding the obtained data.

### 2.5. Statistical Data Analysis

The Statistical Package for Social Sciences (SPSS), version 22.0 (IBM Corp., Armonk, NY, USA) was used for analyzing the data. Categorical variables were described as numbers and percentages. Continuous normally distributed variables were described as means and standard deviations (SDs) or medians and interquartile ranges (IQRs) if the data were not normally distributed. The chi-squared test measured the association between qualitative variables or Fisher’s exact test. Student’s *t*-test was used to compare means and SDs of two sets of normally distributed quantitative data. The Mann–Whitney U test was used if the data were not normally distributed. Statistical significance was set at *p* < 0.05.

Structural equation modeling (SEM) using the AMOS software, version 23, was used to assess the effect of the PACV domains on parental VH as direct, indirect, and total effects of exogenous and endogenous variables for an outcome variable. The goodness of fit of the model was investigated using root mean square residual (RMSR), goodness of fit index (GFI) >0.9, comparative fit index (CFI) >0.9, root mean square error of approximation (RMSEA) <0.05, and normed fit index >0.9. Discriminant analysis was used for predicting the likelihood of outcome categories (acceptance vs. hesitancy) for two or more categories by generating a discriminant function that included several discriminator variables (independent variables).

## 3. Results

### 3.1. Parents’ Characteristics

Most participants were the children’s mothers (68.2%), 40 years or older (59.5%), and had a university degree (53.2%). Approximately 61.8% were not employed, 76.9% had enough income, 57.8% had a family size of 4–5, and 71.1% had no health insurance. Only 15% of participants had a history of chronic disease. Approximately 69.4% had never contracted COVID-19, but a plurality had received the first and second doses or were awaiting the booster dose (45.7%) (Table 1).

### 3.2. Characteristics of Children

Most children were female (64.7%), with a mean age of 12.7 ± 3.6 years, did not have COVID-19 before (91.9%), and had received their compulsory vaccines (97.1%). However, only 4.6% had received an influenza vaccine over the past two years (Table 2).

### 3.3. Proportion of Parents Hesitant about COVID-19 Vaccination

In the study, 81.5% were hesitant to vaccinate their children for COVID-19, and only 18.5% had already done so (Figure 1).

### 3.4. Parents’ and Children’s Characteristics Affecting COVID-19 VH

Mothers were more hesitant regarding children’s vaccination than fathers (88.2% vs. 66.7%, *p* = 0.003). Parents aged <40 years were significantly more hesitant than parents aged >40 years (92.9% vs. 73.8%, respectively, *p* = 0.002). VH was significantly high among illiterate (92%), unemployed (88.8%), and non-health-insured parents (87.8%), *p* < 0.05. Most of the parents who did not receive the COVID-19 vaccine or refused to complete their primary vaccination series (97.2% and 85.7%, respectively) hesitated to vaccinate their children. In contrast, those accepting vaccination had a significantly higher intention to vaccinate their children (72.3% to 100%; *p* = 0.033). Finally, the absence of chronic disease significantly increased VH: 85.7% of parents who did not have chronic disease were hesitant, versus 57.7% of parents who had chronic disease (*p* = 0.001). The mean age of CLD children was significantly lower (11.90 ± 3.473) in the hesitant group than the accepting-vaccination group (16.34 ± 1.4) *p* = 0.001. Previous COVID-19 infection (66.7%) of the children significantly motivated parents to vaccinate them (*p* < 0.0001). The median attitude, trust, and total PACV scores were significantly higher in the hesitant group than in the vaccinated group (*p* = 0.023 and 0.015, respectively) (Table 3).

### 3.5. Structural Equation Model

The framework suggests that the child’s age and family size have a direct effect on parents’ VH. Education level and income had indirect effects on parental VH via the endogenous mediator variable attitude and trust domain for PACV. Low educational level of parents and child sex affected parents’ VH indirectly via the endogenous variable’s safety and efficacy. Moreover, low education affected the attitude and trust domains of PACV. The model had an acceptable goodness of fit (GFI = 0.994, CFI = 1, RMSEA < 0.0001) (Figure 2). In addition to goodness of fit measures, a multi-collinearity test was conducted for all exogenous variables with all endogenous variables (dependent variable and mediator variables) and no inter-correlation was found; all tolerance values were more than 0.2, and VIF (variance inflation factor) values were less than 5, as it is clear in the Appendix A. The Appendix A emphasized that there is no heteroscedasticity; in other words, the variance around the regression line is the same for all values of the predictor variable, so the assumption of homoscedasticity is met.

The direct and indirect effects of these factors are presented in Table 4. Child age had a negative direct effect on parents’ hesitancy: as a child’s age increased by one year, the chance of parental hesitancy decreased by 0.049.

### 3.6. Discriminant Analysis

The discriminant analysis model included safety and efficacy, attitude and trust, child age, and family size. We note the differences between the means of the independent variable for the two categories (already took the COVID-19 vaccine and hesitant) for child age (16.34 ± 1.41 vs. 11.96 ± 3.47), family size (5.50 ± 0.67 vs. 5.07 ± 0.87), attitude and trust (1.56 ± 1.41 vs. 2.34 ± 1.88), and safety and efficacy (6.0 ± 2.39 vs 6.41 ± 2.01). This means that the four variables contribute to the discrimination between VH parents and parents who had already vaccinated their children with CLD for COVID-19. All the discriminator variables were statistically significant for discriminating between the two categories, with a corrected classification of 92.9% (Table 5).

The discriminant functions are presented in Table 4 (Equation: D= −0.017 * safety and efficacy—0.175 * attitude trust + 0.283 * child’s age + 0.310 * family size—4.714).

## 4. Discussion

Many studies have reported high rates of VH in the region toward either the initial series of vaccination [23,24,25,26] or the booster dose [27,28]. However, there is scarcity in the literature regarding the parent’s attitude concerning children’s vaccination, especially among parents of children with chronic diseases. This study investigated the extent of COVID-19 VH and its predictors among parents of children with CLD in Egypt. Overall, 81.5% of parents were hesitant about vaccinating their children. Child age, family size, attitude, and trust in vaccination had direct effects on VH, while income, level of parents’ education, and safety and efficacy of vaccines were indirectly effective.

### 4.1. Vaccination Hesitancy

Parental VH is a serious problem in many countries and will substantially influence the promotion of child vaccination against COVID-19 [22]. We found that nearly four-fifths of parents were hesitant. Having a child treated for CLD may be a contributing factor to this high rate of VH. Few studies assessed VH among parents of children with CLD [33]. A similarly high rate of VH was reported by Alenezi et al. [33]: approximately 54% of 1340 Saudi parents were hesitant to vaccinate their children aged 5–11 years, and 57.2% were unwilling to administer the extra booster vaccination to children aged 12–18 years. Vaccine confidence was associated with acceptance of vaccinating children. Complacency regarding the COVID-19 vaccine was associated with the refusal to vaccinate older children (12–18 years). A low adolescent vaccination rate was reported in Alabama, which provided a catastrophic situation during the winter of 2022, where the spike in the Omicron variant of COVID-19 resulted in a large increase in hospitalizations among young people aged 0–20 years [3]. Orbea et al. [34] reported that 50% of caregivers of hospitalized children were hesitant to vaccinate their children despite believing in the COVID-19 vaccine’s role in controlling the pandemic. They were concerned about the vaccine’s safety and effectiveness. We speculate that disinformation and the politics of COVID-19 vaccination have harmed parents’ intent to vaccinate their children. Consequently, health authorities must provide more health messaging on the safety and effectiveness of vaccination among children through commonly used communication channels. This would combat misinformation and the infodemic and enhance parents trust in vaccination [34].

### 4.2. Determinants of Vaccination Hesitancy

In this study, a range of socioeconomic characteristics, such as child age and sex, parents’ educational level, and income significantly affected parents’ attitudes toward their children’s vaccination. Orbea et al. [34] found that parents who did not intend to vaccinate their children were more likely to be Black (27% vs. 16%, *p* = 0.04) and less likely to be Hispanic/Latin (33% vs. 49%, *p* = 0.02). Furthermore, Derdemezis et al. [35] found that those who were married, had high education and income levels, were nonsmokers, and had stress and depressive symptoms were less likely to be hesitant. Variables related to awareness, knowledge, and trust in authorities regarding COVID-19 were strongly and independently correlated with reduced opposition to existing children’s vaccination programs. This highlights the urgent need for more research in different countries to determine the specific modifiable and non-modifiable risk factors that may affect parents’ attitude towards vaccination. Targeting these resistant groups and modifying their attitude may be considered as an important approach to reduce such high VH rates.

### 4.3. Trust and Attitude

In this study, parents’ trust had a direct negative association to their hesitancy regarding children’s vaccination. For each increase in parents’ trust, the degree of VH decreased by −0.317. In the same way, overall, 55% of parents said that they would refuse to vaccinate their children who were not yet aged eighteen with the COVID-19 vaccine. Parents who trusted the healthcare system had lower odds of vaccine refusal (OR: 0.527, 95% CI: 0.327–0.848) [36]. According to a meta-analysis of COVID-19 vaccine acceptance among pregnant females, older age, trust in COVID-19 vaccines, and fear of COVID-19 were predictors of COVID-19 vaccination uptake, while distrust in the government and concerns regarding the safety and side effects of COVID-19 vaccines were causes of lapsed vaccination in pregnant women. In a previous study, the total rate of COVID-19 vaccination among mothers was found to be 27.5% [37]. It is worth noting that among both pro-vaccine and antivaccine parents, family physicians and other doctors were the most frequently accessed and trusted sources of information. Pediatricians may have a substantial influence on parents’ vaccination decisions for their children [38]. This finding indicates the importance of trust-building in the healthcare system and vaccine delivery to improve uptake in this vulnerable group. Health care providers should be informed about the results of such studies and be actively engaged in all vaccination campaigns. Moreover, they should be aware of rumors and misinformation about vaccination. They can effectively approach populations during their health care center visits due to causes other than vaccination and provide them with information about the role of vaccination to control the pandemic in general and protect their children, specifically.

### 4.4. Safety and Efficacy

The safety and efficacy of the COVID-19 vaccine had an indirect effect on VH through the attitudes and trust of parents. For each increase by 1 in safety and efficacy, VH decreases by 0.036. Due to reports of the limited illness severity among young children, many parents have expressed concern about the safety of the COVID-19 vaccine and questioned the necessity of vaccinating teenagers, prompting them to delay or deny immunization. This finding confirms the need to deliver health messages to the parents of children with CLD. Children with CLD, particularly those with cirrhosis, are at higher risk of developing hepatic decompensation when exposed to vaccines for preventing infections, such as vaccines against hepatitis A and B, pneumococcal illness, influenza, and COVID-19 [39]. Patients with CLD, especially those with cirrhosis, when infected with SARS-CoV-2, have a greater mortality risk from infection with SARS-CoV-2 than those without CLD. The probability of all major adverse events, including pediatric intensive care admission, hospitalization, invasive mechanical ventilation, renal replacement management, and mortality, increases stepwise with the severity of CLD [40]. Defects in adaptive immunity cause patients with CLD to be hyporesponsive to vaccination [39]. Thuluvath et al. [41] found that 76% of CLD patients exhibited sufficient antibody response to COVID-19 vaccinations. In adults, SARS-CoV-2 infection occurred in 15% and 3.7% of 68,048 unvaccinated and 10,441 vaccinated CLD patients with cirrhosis, respectively. For 27,235 completely vaccinated CLD patients without cirrhosis and 8218 fully vaccinated CLD patients with cirrhosis, the estimated incidence rates for breakthrough infections were 5.6 and 5.1 per 1000 person-months, respectively [42]. The previous findings proved the vaccine safety, immunogenicity, and effectiveness in reducing risk of deterioration of CLD, morbidity, and mortality of COVID-19 breakthrough of SARS-CoV-2 among patients with CLD. We speculate these findings may be sufficient to persuade parents to accept vaccination of their children against SARS-CoV-2 infection.

### 4.5. Limitations and Strengths

This study has some limitations that must be considered when interpreting the findings. First, its cross-sectional design restricts its ability to develop a causal link between the studied variables. Therefore, changes in parental VH over time should be studied with a longitudinal approach. Second, the sample was small and was obtained from one setting. Third, due to the simultaneous recording of data on exposure and outcome, the determination of the exposure or the outcome may be influenced by prior knowledge of the condition. Finally, information bias and confounding are among the limitations of this study. However, a systematic random sampling method was adopted to provide all with an equal opportunity for involvement in our study and to overcome bias in sample recruitment as much as possible. A multicenter study with a larger sample is warranted. Despite these limitations, this study makes a significant contribution. To the best of our knowledge, this is one of the first studies to evaluate VH among parents of children with CLD and to explore the factors that can influence parents’ attitudes toward COVID-19 vaccines. Advanced statistical analyses, including SEM and discriminant analysis, were used to obtain more insight into VH among parents. SEM was used instead of a traditional regression model to identify both direct and indirect predictors of VH. Discriminant analysis was used to assess the large distance between centroids for increased efficiency in distinguishing the different categories of outcome variables.

## 5. Conclusions

The VH rate among the parents of children with CLD was remarkably high. Different modifiable and nonmodifiable socioeconomic factors had direct (child age and family size) and indirect effects (education level and income) on parents’ attitudes toward COVID-19 vaccination. More attention should be paid to informing caregivers of the crucial benefits of vaccination, targeting this vulnerable population to reduce complications of SARS-CoV-2 infection. Health messages addressing the safety, immunogenicity, and effectiveness of COVID-19 vaccine among children with chronic diseases should be continually delivered through the commonly used communication channels with effective engagement of health care professionals. This study emphasizes the value of proactive public health and policy to limit pandemic impacts on children’s health, especially those with chronic diseases. Therefore, successful vaccination may reduce the significant social and economic burden of COVID-19 on children and their parents.

## Figures and Tables

**Figure 1 vaccines-10-02094-f001:**
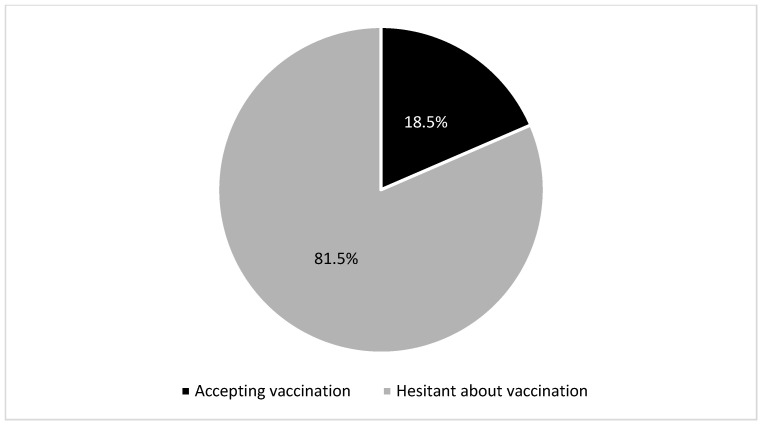
Attitudes of parents of patients with chronic liver disease toward the COVID-19 vaccine.

**Figure 2 vaccines-10-02094-f002:**
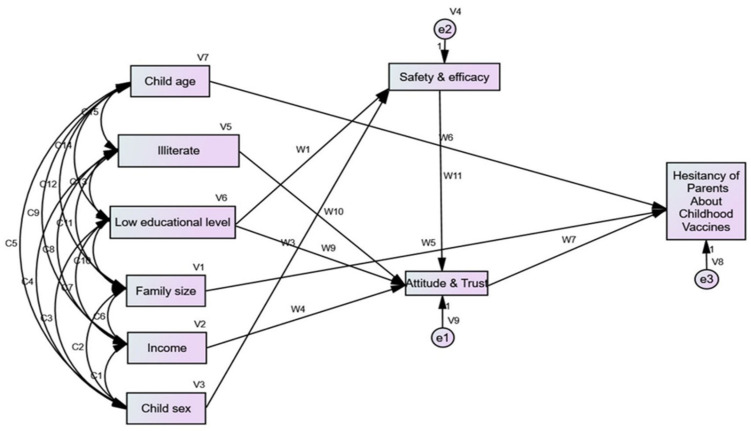
Visualization of the structural equations model for Hesitancy of Parents about Childhood Vaccines.

**Table 1 vaccines-10-02094-t001:** Characteristics and COVID-19 related data of parents (*n* = 173).

Variable	Category	N (%)
Gender	Female	118 (68.2%)
	Male	55 (31.8%)
Age	<40 years	70 (40.5%)
	≥40 years	103 (59.5%)
Relation to the child	Father	48 (27.7%)
	Mother	118 (68.2%)
	Others	5 (3.5%)
Level of education	Illiterate	25 (14.5%)
	Low educated ^1^	56 (32.4%)
	Medium to high educated ^2^	92 (53.2%)
Workplace	Not employed	107 (61.8%)
	Government	47 (27.2%)
	Private	19 (11.0%)
Work sector	Health	6 (3.5%)
	Non-Health	167 (96.5%)
Health insurance	No	123 (71.1%)
	Private	6 (3.5%)
	Governmental	44 (25.4%)
Income	Not enough, on a loan and cannot pay back	4 (2.3%)
	Not enough, on a loan but can pay back	14 (8.1%)
	Enough	133 (76.9%)
	Enough and saving	22 (12.7%)
Older adults living in the same home	No	111 (64.2%)
	Yes	62 (35.8%)
Family size	2–3	6 (3.5%)
	4–5	100 (57.8%)
	>5	67 (38.7%)
Previous COVID-19 infection	No	120 (69.4%)
	Not Sure	7 (4.0%)
	Yes	46 (26.6%)
COVID-19 vaccine status	Does not want to take the vaccine	36 (20.8%)
	Took the first dose and is awaiting the second	22 (12.7%)
	Took the first dose but does not want to take the second dose	7 (4.0%)
	Took the first and second doses and is awaiting the booster dose	79 (45.7%)
	Took the first and second doses but did not want to take the booster dose	4 (2.3%)
	Took the three doses	18 (10.4%)
	Wants to take the vaccine, but it is not scheduled yet	7 (4.0%)
Chronically diseased parents	Yes	26 (15.0%)

^1^ Write and read, primary education, and preparatory education. ^2^ High school, university education, and postgraduate education.

**Table 2 vaccines-10-02094-t002:** Sociodemographic and COVID-19 data for children with chronic liver disease (*n* = 173).

Variable		N (%)
Sex	Female	112 (64.7%)
	Male	61 (35.3%)
Age	Mean ± SD	12.77 ± 3.6
Previous COVID-19 infection	No	159 (91.9%)
	Not sure	5 (2.9%)
	Yes	9 (3%)
The child received the scheduled vaccine	Yes	168 (97.1%)
The child received the influenza vaccine last 2 years	Yes	8 (4.6%)
Parents’ intentions to allow COVID-19 vaccination	Already took the COVID vaccine	32 (18.5%)
	Hesitant	141 (81.5%)

**Table 3 vaccines-10-02094-t003:** Sociodemographic characteristics across vaccinated children.

Variable		Category	Parents’ Intention		Chi-Square Test	df	*p.*
			Accepting Vaccination	Hesitant			
	Gender	Female	16 (13.6%)	102 (86.4%)	6.003	1	0.014 *
		Male	16 (29.1%)	39 (70.9%)			
	Age	<40 years	5 (7.1%)	65 (92.9%)	10.054	1	0.002 *
		>40 years	27 (26.2%)	76 (73.8%)			
	Relation to the child	Father	16 (33.3%)	32 (66.7%)	10.797	1	0.001 *
		Mother	14 (11.8%)	105 (88.2%)			
	Workplace	Not employed	12 (11.2%)	95 (88.8%)	10.147	2	0.006 *
		Government	15 (31.9%)	32 (68.1%)			
		Private	5 (26.3%)	14 (73.7%)			
	Health insurance	No	15 (12.2%)	108 (87.8%)	11.567 ^#^	2	0.003 *
		Private	1 (16.7%)	5 (83.3%)			
		Governmental	16 (36.4%)	28 (63.6%)			
	Older adults living in the same home	No	18 (16.2%)	93 (83.8%)	1.069	1	0.301
		Yes	14 (22.6%)	48 (77.4%)			
	Previous COVID-19 infection	No	17 (14.2%)	103 (85.8%)		2	0.05
		Not Sure	3 (42.9%)	4 (57.1%)	6.006		
		Yes	12 (26.1%)	34 (73.9%)			
	COVID-19 vaccine status	Does not want to take the vaccine	1 (2.8%)	35 (97.2%)	12.448 ^#^	6	0.033 *
		Took the first dose and is awaiting the second	6 (27.3%)	16 (72.7%)			
		Took the first dose but does not want to take the second dose	1 (14.3%)	6 (85.7%)			
		Took the first and second doses and is awaiting the booster dose	20 (25.3%)	59 (74.7%)			
Parents’ characteristics		Took the first and second but does not want to take the booster dose	-	4 (100%)			
		Took the three doses	4 (22.2%)	14 (77.8%)			
		Wants to take the vaccine, but it is not scheduled yet	-	7 (100%)			
	Chronic diseased parents	No	21 (14.3%)	126 (85.7%)	11.507	1	0.001 *
		Yes	11 (42.3%)	15 (57.7%)			
	Child age	Mean ± SD	16.34 ± 1.4	11.90 ± 3.473	11.41 ^$^		0.001 *
	Child sex	Female	22 (19.6%)	90 (80.4%)	0.277	1	0.599
		Male	10 (16.4%)	51 (83.6%)			
Children’s characteristics	Child with a previous COVID-19 infection	No	23 (14.5%)	136 (85.5%)	16.884 ^#^	2	0.0001 *
		Not sure	3 (60.0%)	2 (40.0%)			
		Yes	6 (66.7%)	3 (33.3%)			
Parents’ Attitudes about Childhood Vaccines (PACV)	Behavior	Median (IQR)	0 (0–0)	0 (0–0)	0.83 ^$$^		0.407
	Safety and efficacy	Median (IQR)	7 (6.25–7)	7 (7–7)	0.754 ^$$^		0.451
	Attitude and trust	Median (IQR)	1 (0.25–2)	2 (1–3.5)	2.145 ^$$^		0.032 *
	Total PACV Score ^##^	Median (IQR)	26.67 (23.3–33.3)	30 (26.67–33.33)	2.424 ^$$^		0.015 *

df: degree of freedom, ^#^ Fisher’s Exact Test, ^$^ Independent Samples *t*-test, ^$$^ Mann–Whitney U test, * Significant, ^##^ PACV total score = 100.

**Table 4 vaccines-10-02094-t004:** Value and direction of the total, direct, and indirect effect of exogenous (indirect) endogenous (direct) variables.

Variable	Total Effect and Direction	Direct Effect and Direction	Indirect Effect and Direction
Child age	−0.049	−0.049	0
Level illiterate	−0.018	0	−0.018
Level low	−0.014	0	−0.014
Family size	−0.053	−0.053	0
Income	−0.048	0	−0.048
Child sex	−0.004	0	−0.004
Safety and efficacy	0.004	0	0.004
Attitude trust	0.030	0.030	0

**Table 5 vaccines-10-02094-t005:** Discriminant analysis and standardized canonical discriminant function coefficients.

Parents’ Hesitancy about Vaccines	Accepting Vaccination Mean ± SD	Hesitant Mean ± SD	Wilks’ Lambda	Function	*p.*
Safety and efficacy	6.0 ± 2.39	6.41 ± 2.01	0.98	−0.036	0.092
Attitude and trust	1.56 ± 1.41	2.34 ± 1.88	0.96	−0.317	0.028
Child age	16.34 ± 1.41	11.96 ± 3.47	0.78	0.904	0.000
Family size	5.50 ± 0.67	5.07 ± 0.87	0.96	0.259	0.009

## Data Availability

Data are available in Appendix A.

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
