# Peer review of "COVID-19 Vaccine Hesitancy among Parents of Children with Chronic Liver Diseases"

_vaccines, 2022, doi:10.3390/vaccines10122094_

Round 1
Reviewer 1 Report
Dear Authors
It was with great pleasure that I reviewed your manuscript.
The main issue addressed by this research is the hesitation that parents of children with liver disease show toward vaccination. In this aspect the manuscript is quite explicit.
The topic is relevant but not very original. The only originality is that it concerns parents of children with liver disease.
What adds to other studies is that these children train liver pathologies.
The methodology applied was the one indicated for this type of study.
The main problem with this study is that the discussion should be more in-depth. The researchers do not present any limitations of the study. They also do not present practical implications. Finally, they also need to present the conclusions chapter.
The references are adequate. The tables and figures are well presented. In this commentary, the general structural equation model is included.
My Best Regards
Author Response
Reviewer #1 Comments and Suggestions for Authors
1-The main problem with this study is that the discussion should be more in-depth.
To overcome, we added some paragraphs to go more in depth:
|
They were concerned about the vaccine’s safety and effectiveness. We speculate that dis-information and the politics of COVID-19 vaccination have harmed parents' intent to vaccinate their children. Consequently, health authorities must provide more health message on the safety and effectiveness of vaccination among children through commonly used communication channels. This would combat misinformation and infodemic and enhance parents trust in vaccination [35] |
Vaccine hesitancy section p. 10 |
Line 257 -262 |
|
This highlights the urged need for more research in different countries to determine the specific modifiable and non-modifiable risk factors that may affect parents’ attitude to-wards vaccination. Targeting these resistant groups and modifying their attitude may be considered as an important approach to reduce such high VH rates. |
Determinants of vaccination hesitancy section p.10-11 |
Line 272-276 |
|
This finding indicates the importance of trust-building in the healthcare system and vac-cine delivery to improve uptake in this vulnerable group. Health care providers should be informed about the results of such studies and be actively engaged in all vaccination campaigns. Moreover, they should know about rumors and misinformation about vaccination. They can effectively approach population during their health care centres visits due causes other than vaccination and provide them with information about the role of vaccination to control the pandemic in general and protect their children specifically. |
Trust and attitude section p.10-11
|
Line 291-298 |
|
The previous findings proved the vaccine safety, immunogenicity, and effectiveness in reducing risk of deterioration of CLD, morbidity and mortality of COVID-19, breakthrough of SARS-CoV-2 among patients with CLD. We speculate these findings may be sufficient to persuade parents to accept vaccination of their children against SARS-CoV-2 infection. |
Safety and efficacy section p.11 |
Line 320-324 |
2-The researchers do not present any limitations of the study. They also do not present practical implications. Finally, they also need to present the conclusions chapter.
To improve our manuscript, we added the following:
|
This study has some limitations that must be considered when interpreting the findings. First, its cross-sectional design restricts its ability to develop a causal link between the studied variables. Therefore, changes in parental VH over time should be studied with a longitudinal approach. Second, the sample was small and was obtained from one setting. Third, due to the simultaneous recording of data on exposure and outcome, the de-termination of the exposure or the outcome may be influenced by prior knowledge of the condition. Finally, information bias and confounding among the limitations of this study. |
Limitation & strengths section p.12 |
Line 326-332 |
|
Discriminant analysis was used to assess the large distance between centroids for in-creased efficiency in distinguishing the different categories of outcome variables. |
Limitation & strengths section p.12 |
Line 341-343 |
|
Health messages addressing the safety, immunogenicity, and effectiveness of COVID-19 vaccine among children with chronic diseases should be continuously delivered through the commonly used communication channels with effective engagement of health care professional |
Conclusion section p. 12 |
Line 350-353 |

Reviewer 2 Report
The topic of the manuscript is within the scope of the Journal and could be valuable to the scientific audience. The quality of the research design is acceptable.
The problem question is what are peculiarities of COVID-19 vaccine hesitancy and its predictors among parents of children with chronic liver disease in Egypt?
The topic of the manuscript is relatively original and is within the scope of the Journal and could be valuable to the scientific audience. This study investigated the extent of COVID-19 vaccine hesitancy (VH) and its predictors in parents of children with chronic liver disease (CLD) in Egypt (in specific country).
This study contributes to the ongoing arguments related to the extent of COVID-19 vaccine hesitancy among parents of children with chronic liver diseases in Egypt (in specific country).
But, the specific improvements should the authors consider regarding the methodology: INTRODUCTION The research hypothesis has been not formulated. METHOD Please address the following issues: Have the assumptions for Structural equation modelling (SEM) been tested and met? For instance, no multicollinearity, homoscedasticity. RESULTS Some clarifications are however needed. The technique of data analyses seems appropriate but the results section is missing some important information. My suggestion is that all statistical notations for analyses should be written (not only p-values but also statistical analysis test values (for instance, chi-square tests’ values, Student’s t-test values, Mann-Whitney U test values), degrees of freedom) because that is necessary for the proper interpretation of results.
Conclusions are consistent with the evidence and arguments presented, however, COVID-19 VH predictors among parents of children with CLD in Egypt must be included in conclusions part.
The references are appropriate.
My suggestion on the Table 3 is that all statistical notations for analyses should be included (not only p-values but also statistical analysis test values (for instance, chi-square tests’ values, Student’s t-test values, Mann-Whitney U test values), degrees of freedom) because that is necessary for the proper interpretation of results. Figures are appropriate.
TO SUM UP I think the author(s) need to make the recommended corrections.
Author Response
Reviewer #2 Comments and Suggestions for Authors
1-INTRODUCTION The research hypothesis has been not formulated.
|
Many studies have reported high rates of VH in the region towards either the initial series of vaccination [23]–[26] or the booster dose [27], [28]. However, the is scarcity in literature on the parent’s attitude about children vaccination especially among parents of children with chronic diseases. Having a child with a chronic disease may negatively affect parental decision-making regarding vaccination. So that, this study aimed to investigate COVID-19 VH and its predictors among parents of children with CLD in Egypt. |
Introduction section p. 1-2 |
Line 100-105 |
2-METHOD, please address the following issues: Have the assumptions for Structural equation modelling (SEM) been tested and met? For instance, no multicollinearity, homoscedasticity.
Yes, it is well clarified in result section and additional supplementary table 1 and figures 1,2
|
In addition to goodness of fit measures a multi-collinearity test was done for all exogenous variables with all endogenous variables (dependent variable and mediator variables) and no inter-correlation was found where all tolerance values were more than 0.2, also VIF (Variance Inflation Factor) were less than 5, as it is clear in the Supplementary table 1. The Supplementary figure 1 and 2 emphasized that there is no heteroscdecity, on other words the variance around the regression line is the same for all values of the predictor variable so the assumption of homoscedasticity is met. |
Result section p.8 |
Line 203-210 |
3-RESULTS Some clarifications are however needed. The technique of data analyses seems appropriate, but the results section is missing some important information. My suggestion is that all statistical notations for analyses should be written (not only p-values but also statistical analysis test values (for instance, chi-square tests’ values, Student’s t-test values, Mann-Whitney U test values), degrees of freedom) because that is necessary for the proper interpretation of results.
We added all needed data to table 3 for more clarifications as follows:
Table 3. Sociodemographic characteristics across vaccinated children.
|
Variables |
Category |
Parents intentions |
Chi Square Test |
df |
P |
||
|
Accepting vaccination |
Hesitant |
||||||
|
Parents characteristics |
Gender |
Female |
16 (13.6%) |
102 (86.4%) |
6.003 |
1 |
0.014* |
|
Male |
16 (29.1%) |
39 (70.9%) |
|||||
|
Age |
<40 year |
5 (7.1%) |
65 (92.9%) |
10.054 |
1 |
0.002* |
|
|
>40 year |
27 (26.2%) |
76 (73.8%) |
|||||
|
Relation to the child |
Father |
16 (33.3%) |
32 (66.7%) |
10.797 |
1 |
0.001* |
|
|
Mother |
14 (11.8%) |
105 (88.2%) |
|||||
|
Workplace |
Not employed |
12 (11.2%) |
95 (88.8%) |
10.147 |
2 |
0.006* |
|
|
Government |
15 (31.9%) |
32 (68.1%) |
|||||
|
Private |
5 (26.3%) |
14 (73.7%) |
|||||
|
Health insurance |
No |
15 (12.2%) |
108 (87.8%) |
11.567#
|
2 |
0.003* |
|
|
Private |
1 (16.7%) |
5 (83.3%) |
|||||
|
Governmental |
16 (36.4%) |
28 (63.6%) |
|||||
|
Older adults living in the same home |
No |
18 (16.2%) |
93 (83.8%) |
1.069 |
1 |
0.301 |
|
|
Yes |
14 (22.6%) |
48 (77.4%) |
|||||
|
Previous COVID-19 infection |
No |
17 (14.2%) |
103 (85.8%) |
6.006 |
2 |
0.05 |
|
|
Not Sure |
3 (42.9%) |
4 (57.1%) |
|||||
|
Yes |
12 (26.1%) |
34 (73.9%) |
|||||
|
COVID-19 vaccine status |
Does not want to take the vaccine |
1 (2.8%) |
35 (97.2%) |
12.448# |
6 |
0.033* |
|
|
Took the first dose and is awaiting the second |
6 (27.3%) |
16 (72.7%) |
|||||
|
Took the first dose but does not want to take the second dose |
1 (14.3%) |
6 (85.7%) |
|||||
|
Took the first and second doses and is awaiting the booster dose |
20 (25.3%) |
59 (74.7%) |
|||||
|
Took the first and second but does not want to take the booster dose |
- |
4 (100%) |
|||||
|
Took the three doses |
4 (22.2%) |
14 (77.8%) |
|||||
|
Wants to take the vaccine, but it is not scheduled yet |
- |
7 (100%) |
|||||
|
Chronic diseased parents |
No |
21 (14.3%) |
126 (85.7%) |
11.507 |
1 |
0.001* |
|
|
Yes |
11 (42.3%) |
15 (57.7%) |
|||||
|
Children characteristics |
Child age |
Mean ± SD |
16.34 ± 1.4 |
11.90 ± 3.473 |
t=11.41 |
|
0.001* |
|
Child sex |
Female |
22 (19.6%) |
90 (80.4%) |
0.277 |
1 |
0.599 |
|
|
Male |
10 (16.4%) |
51 (83.6%) |
|||||
|
Child with a previous COVID-19 infection |
No |
23 (14.5%) |
136 (85.5%) |
16.884# |
2 |
0.0001* |
|
|
Not sure |
3 (60.0%) |
2 (40.0%) |
|||||
|
Yes |
6 (66.7%) |
3 (33.3%) |
|||||
|
Parent Attitudes about Childhood Vaccines (PACV) |
PACV |
|
|
||||
|
Behavior |
Median (IQR) |
0 (0–0) |
0 (0–0) |
U=0.83 |
|
0.407 |
|
|
Safety and efficacy |
Median (IQR) |
7 (6.25–7) |
7 (7–7) |
U=0.754 |
|
0.451 |
|
|
Attitude and trust |
Median (IQR) |
1 (0.25–2) |
2 (1–3.5) |
U=2.145 |
|
0.032* |
|
|
Total PACV Score |
Median (IQR) N.B Full score 100 |
26.67(23.3–33.3) |
30 (26.67–33.33) |
U=2.424 |
|
0.015* |
|
|
df degree of freedom, # Fisher’s Exact Test, t Independent Samples t test, U Mann-whitney test, *Significant |
|||||||
4-Conclusions are consistent with the evidence and arguments presented, however, COVID-19 VH predictors among parents of children with CLD in Egypt must be included in conclusions part.
|
The VH rate among the parents of children with CLD was remarkably high. Different modifiable and nonmodifiable socioeconomic factors had direct (Child age and Family size) and indirect effects (Education level and income) on parents’ attitudes toward COVID-19 vaccination. |
Conclusion section p. 12 |
Line 345-348 |
|
Health messages addressing the safety, immunogenicity, and effectiveness of COVID-19 vaccine among children with chronic diseases should be continuously delivered through the commonly used communication channels with effective engagement of health care professional |
Conclusion section p. 12 |
Line 350-353 |

Round 2
Reviewer 1 Report
Dear Authors
I would like to congratulate you on all the corrections you have made to the manuscript. It has become much better.
My Best Regards